# Practical Nonlinear Model Predictive Controller Design for Trajectory Tracking of Unmanned Vehicles

**Hui Pang** [1] , **Minhao Liu** [1] , **Chuan Hu** [2],*** and Nan Liu** [1]

1   School of Mechanical and Precision Instrument Engineering, Xi'an University of Technology, Xi'an 710048, China; panghui@xaut.edu.cn (H.P.); 2200220079@stu.xaut.edu.cn (M.L.); 2200220062@stu.xaut.edu.cn (N.L.)
2   Department of Mechanical Engineering, University of Alaska Fairbanks, Fairbanks, AK 99775, USA
*   Correspondence: chuan.hu.2013@gmail.com

**Abstract:** The trajectory tracking issue of unmanned vehicles has attracted much attention recently, with the rapid development and implementation of sensing, communication, and computing technologies. This paper proposes a nonlinear model predictive controller (NMPC) for the trajectory tracking application of an unmanned vehicle (UV). First, a two-degree-of-freedom (2-DOF) kinematics model of this UV is used to derive the desirable controller with two control variables as forward velocity and yaw angle. Next, the one-step Euler method is employed to establish the nonlinear prediction model, then a nonlinear optimization objective function is formulated to minimize the tracking errors of forward velocity and yaw angle from a preset time-varying reference road. Finally, the effectiveness of the proposed NMPC scheme is assessed under two different driving scenarios via MATLAB simulations. The simulation results confirm that the proposed NMPC scheme reveals better control accuracy and computational efficiency than the standard MPC controller under two different prescribed roads. Moreover, an outdoor field test is conducted to verify the performance of the proposed NMPC scheme, and the results show that the proposed NMPC can be applied to the real vehicle and can improve the tracking accuracy and the driving stability of trajectory tracking.

**Keywords:** unmanned vehicle; nonlinear model prediction controller; trajectory tracking; outdoor field test

## 1. Introduction

Currently, the driverless technology of unmanned vehicles (UVs) has attracted more and more attention from industrial and academic circles because the efficient control schemes of UVs have great potential in guaranteeing vehicle safety performance and traffic efficiency [1,2]. Trajectory tracking control [3–5] is a key link connecting environment perception and motion control for the implementation of UVs, and efficient trajectory tracking can improve the safety performances and ride comforts of UVs.

In the current research on the trajectory tracking problems, many intelligent optimization algorithms have been widely proposed for the trajectory tracking of UVs, such as the ant colony algorithm [6], particle swarm optimization algorithm [7], genetic algorithm [8], graphic algorithm [9,10], etc. Although these algorithms can track the prescribed path, they mainly adopt the infinite iteration method that requires a large amount of computation to obtain the optimal solutions; therefore, the real-time performance and tracking accuracy of those methods may not be guaranteed. Herein, as a popular iterative optimized control approach, model predictive control (MPC) [11,12] can simultaneously incorporate system state constraints and control input constraints into the controller design process, i.e., at each sample time, the future system inputs and outputs can be obtained by updating the control plant and can be optimized via an appropriate optimization algorithm in the predictive horizon, wherein the system constraints are easily put in an explicit form.

As reported in the literature [13–20], the MPC method has been widely employed in the trajectory tracking controller development for different types of UVs. For instance, in the literature [18], a control plant integrating the kinematic motions and the dynamic behaviors of an automated guided vehicle was established, and an MPC-based trajectory tracking controller was then designed. In [19], an MPC method was presented to control the forward steering of autonomous vehicles by continuously linearizing nonlinear vehicle models. In addition, a direct data-driven MPC method has been proposed in [20] to relax the laborious model identification procedure. A linear-parameter-varying MPC approach was proposed in [21] for UVs, and the control scheme was validated through simulation and experiment investigations. Similarly, in order to estimate the driver intentions and the underlying behaviors, an MPC approach integrating with recurrent neural network and memory cell was proposed for an unmanned vehicle [22], and the simulation results verified the improvements in trajectory tracking performances for this vehicle.

In short, most of the early literature [18–20,22] mainly focuses on the simulation investigations on trajectory tracking problems, and the real-platform or field test study is lacking. Moreover, due to the MPC execution load and extra modeling errors, it is hard to linearize the original (nonlinear) system around the current working point and then design an effective model-based predictive controller.

Fortunately, a nonlinear model predictive control (NMPC) approach has been extensively adopted in the practical path planning and trajectory tracking problem due to its inherent profits to deal with the nonlinear input constraints aiming at speeding up the online computation. In [23], an NMPC approach was presented to address the safety issues regarding collision avoidance and lateral stability of unmanned ground vehicles in high-speed conditions. Further, a trajectory tracking NMPC strategy was proposed in [24] to address the explicit state and input constraints for autonomous surface craft, and the real-time implementation of the NMPC was validated through the experimental results. In [25], a distributed control scheme was provided to achieve the accurate tracking control of the autonomous underwater vehicle motion by using NMPC techniques. It can be concluded from these two studies that the NMPC algorithm is very appropriate for solving the nonlinear optimization problem with lower computational cost. Besides, it is inevitably encountered with the regular and irregular roads in the real world. With the NMPC method, the computational efficiency and control accuracy for a nonlinear control system can be guaranteed [26].

Therefore, inspired by the literature mentioned above, this paper proposes a practical NMPC controller design for the trajectory tracking application of a UV considering the nonlinear road trajectory. The main contributions of this work are summarized as:

(1) A novel NMPC controller is proposed to achieve the accurate trajectory tracking of the UV under different prescribed roads, wherein the one-step Euler method is used to establish the nonlinear prediction model. As model predictive control is an iterative process, the Euler method has the advantages of a wide range of numerical solutions, a simple form that is easy to calculate, thus the tracking errors of forward velocity and yaw angle are minimized through a nonlinear optimization method;

(2) MATLAB simulations are carried out to verify the control performances of the proposed NMPC controller under two different driving scenarios, and the results show that the strategy can deal with the nonlinear road trajectory well, and can improve the tracking accuracy and the driving stability;

(3) A simple test platform consisting of a scaled-down real racing car, sensors, microcontroller, and host computer is built up to verify the effectiveness of the designed NMPC controller in UV trajectory tracking applications.

The rest of this article is arranged as follows. The UV's kinematic model formulation is described in Section 2. Next, the design procedure of this expected NMPC controller is provided in Section 3. Then, in order to verify the control performance of the presented NMPC controller, both simulation and field test verifications are orderly conducted and

discussed in Sections 4 and 5. Finally, the conclusions and perspectives of this paper are presented in Section 6.

## 2. The Unmanned Vehicle's Kinematics Model

Here, a kinematic model of UV with two degree-of-freedoms (2-DOF) is adopted to perform the controller synthesis [21]. To mimic a real unmanned vehicle's kinematic behaviors, it is assumed that this UV is driven by a servo motor installed in a rear wheel, and is steered by a servo motor installed in a front wheel. By ignoring the side-slip angle when making the front-wheel steering operation, as well as considering the longitudinal speed to be a constant value, one can construct the 2-DOF kinematic model as shown in Figure 1, in which point $A(X_f, Y_f)$ and $B(X_r, Y_r)$ stand for the center positions of the front axle and rear axle, and the other symbols used in this UV kinematic model are listed in Table 1.

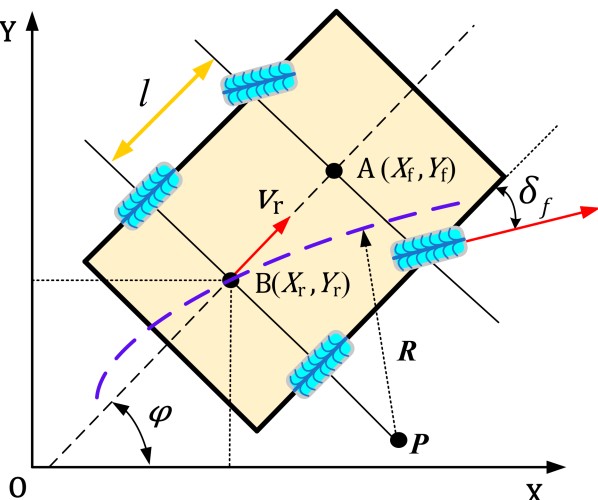

**Figure 1.** The 2-DOF kinematic model of UV.

**Table 1.** The used symbols and parameters of UV.

| Symbol | Description | Symbol | Description |
|---|---|---|---|
| $A(X_f, Y_f)$ | The center of the front axle | $l$ | The distance from point A to B |
| $B(X_r, Y_r)$ | The center of the rear axle | $\omega$ | The yaw rate of UV |
| $\delta_f$ | The deflection angle of the front axle | $\varphi$ | The yaw angle of UV |
| $v_r$ | The forward speed of point B | | |

In terms of the kinematic relationships of this UV, $v_r$ can be expressed by

$$v_r = \dot{X}_r \cos\varphi + \dot{Y}_r \sin\varphi, \tag{1}$$

The kinematic constraints between the rear- and front- axles of this UV are easily obtained

$$\dot{X}_r \sin\varphi = \dot{Y}_r \cos\varphi, \tag{2}$$

$$\dot{X}_f \sin(\varphi + \delta_f) = \dot{Y}_f \cos(\varphi + \delta_f), \tag{3}$$

By integrating Equations (1) and (2), one obtains

$$\begin{cases} \dot{X}_r = v_r \cos\varphi \\ \dot{Y}_r = v_r \sin\varphi \end{cases}, \tag{4}$$

By further transformation, we have the geometric relationship between points A and B satisfying

$$\begin{cases} X_f = X_r + l\cos\varphi \\ Y_f = Y_r + l\cos\varphi \end{cases}, \tag{5}$$

For simplicity, let $\omega$ denote the derivative of $\varphi$, and by substituting Equation (4) into the derivative of Equation (5), we have

$$\begin{cases} \dot{X}_f = v_r\cos\varphi - l\dot{\varphi}\sin\varphi \\ \dot{Y}_r = v_r\sin\varphi + l\dot{\varphi}\cos\varphi \end{cases}. \tag{6}$$

By further integrating Equation (6) with Equation (3), we obtain

$$\dot{\varphi} = \omega = \frac{v_r}{l}\tan\delta_f, \tag{7}$$

The expressions for $R$ and $\delta_f$ can be obtained as follows:

$$\begin{cases} R = \frac{v_r}{\omega} \\ \delta_f = \arctan\left(\frac{l}{R}\right) \end{cases}, \tag{8}$$

By combining Equations (4) and (7), we have

$$\begin{bmatrix} \dot{X}_r \\ \dot{Y}_r \\ \dot{\varphi} \end{bmatrix} = \begin{bmatrix} \cos\varphi \\ \sin\varphi \\ \tan\delta_f/l \end{bmatrix} v_r, \tag{9}$$

Thus, the state-space form of the kinematic Equation for this UV can be formulated as

$$\begin{bmatrix} \dot{X}_r \\ \dot{Y}_r \\ \dot{\varphi} \end{bmatrix} = \begin{bmatrix} \cos\varphi & 0 \\ \sin\varphi & 0 \\ 0 & 1 \end{bmatrix} \begin{bmatrix} v_r \\ \omega \end{bmatrix}. \tag{10}$$

Define $\boldsymbol{\zeta}_{kout} = \begin{bmatrix} X_r & Y_r & \varphi \end{bmatrix}^T$ as the system output vector and $\boldsymbol{u}_{kin} = \begin{bmatrix} v_r & \omega \end{bmatrix}^T$ as the control input vector, then Equation (10) is further rewritten as

$$\dot{\boldsymbol{\zeta}}_{kout} = \begin{bmatrix} \cos\varphi & 0 \\ \sin\varphi & 0 \\ 0 & 1 \end{bmatrix} \boldsymbol{u}_{kin}. \tag{11}$$

It should be noted that the kinematics model can represent the relationship between the vehicle's state of motion and the control input, thus the model predictive controller can achieve the purpose of predetermined control. Now, the kinematics model of the UV is completed, and the proposed controller design will be illustrated in the following.

## 3. Design of Nonlinear Model Predictive Control

As a popular iterative optimization method, model predictive control has been extensively applied to the trajectory tracking and path-following control of UVs in recent years. For a common MPC, it consists predictive model, rolling optimization, and feedback correction. It should be noticed that a predictive model can predict the future system inputs and outputs according to the current system states, rolling optimization can generate the optimized control sequence, and feedback correction is usually to feed the system states at a certain current time back to the control system as the new input to perform the optimization calculation at the next iteration. Particularly, the NMPC can actually handle the linear and nonlinear trajectory tracking problems with the upper-lower limits, herein the one-step Euler method is used to derive the nonlinear model prediction controller with expecting

to enhance the tracking accuracy and rate of the UV. The proposed NMPC framework is shown in Figure 2.

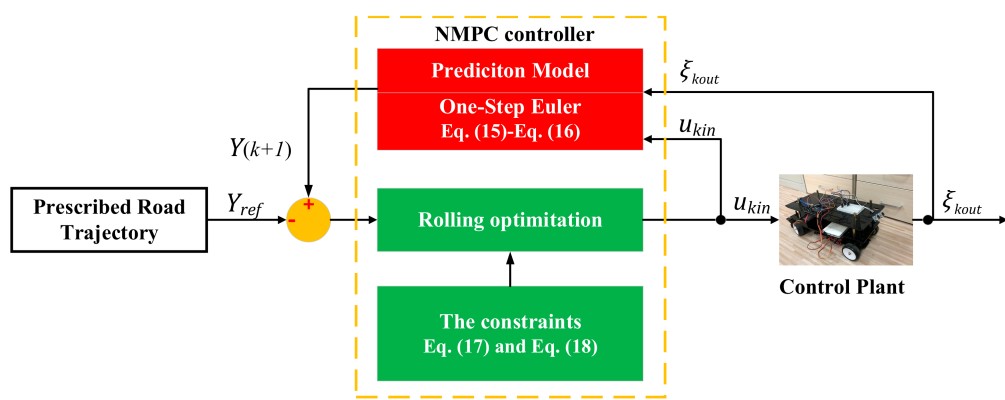

**Figure 2.** Block diagram of proposed NMPC controller.

### 3.1. Establishment of Nonlinear Prediction Model

To facilitate the expected nonlinear prediction model establishment for this UV, Equation (11) can be reconfigured as a nonlinear function expression as

$$\begin{cases} \dot{x} = f(x(t), u(t)) \\ y = g(x(t), u(t)) \end{cases}.$$ (12)

Define $x = [\varphi, X_{\mathrm{r}}, Y_{\mathrm{r}}]^T$ as the state vector, i.e., $\xi_{kout}$, $u(t)$ as the control input vector, i.e., $u_{kin}$, thus we can have

$$f(x(t), u(t)) = \begin{bmatrix} \omega \\ v_{\mathrm{r}} \cos \varphi \\ v_{\mathrm{r}} \sin \varphi \end{bmatrix},$$ (13)

$$g(x(t), u(t)) = \begin{bmatrix} 0 & 1 & 0 \\ 0 & 0 & 1 \end{bmatrix} x(t).$$ (14)

In order to represent the NMPC controller design as a convex optimization problem, the discretization form of Equation (12) can be derived as

$$\begin{cases} x(k+1) = F(x(k), x(k)) \\ y(k) = G(x(k)) \end{cases}.$$ (15)

where $F(x(k), x(k))$ and $G(x(k))$ represent the discretized form of $f(x(t), u(t))$, and $g(x(t), u(t))$.

Furthermore, by introducing the sampling time $T$, the one-step Euler method is employed to describe the prediction model of Equation (15) as follows

$$\begin{cases} x(k+1) = x(k) + \begin{bmatrix} \omega(k) \\ v_{\mathrm{r}}(k) \cos(\varphi(k)) \\ v_{\mathrm{r}}(k) \sin(\varphi(k)) \end{bmatrix} T = x(k) + f(x(k), u(k)) T \\ y(k) = g(x(k), u(k)) \end{cases}.$$ (16)

**Remark 1.** *If the sampling time T is too short, the calculation burden will increase and the tracking accuracy of the control system will be affected. In contrast, if T is too large, a large cumulative error will inevitably be encountered in the control process, which will impose negative impacts on the dynamical performances of the control system. According to the Shannon sampling theory and the control periods of the actuators, the sample time is selected as T = 0.02 s in this paper.*

For the control input $\boldsymbol{u}(k)$, it is necessary to introduce its physical constraints, which is expressed as follows:

$$\boldsymbol{u}_{min}(k) \leq \boldsymbol{u}(k) \leq \boldsymbol{u}_{max}(k), \tag{17}$$

Note that $\boldsymbol{y}_{ref}$ is the reference road trajectory, $\Delta \boldsymbol{y}_{min}$ and $\Delta \boldsymbol{y}_{max}$ represent for the minimal and maximal errors of the deviation between $\boldsymbol{y}(k)$ and $\boldsymbol{y}_{ref}$, both of them can be adjusted in the process of rolling optimization. The constraint relationship between them is expressed as follows:

$$\Delta \boldsymbol{y}_{min} \leq \boldsymbol{y} - \boldsymbol{y}_{ref} \leq \Delta \boldsymbol{y}_{max}. \tag{18}$$

To make a clear distinguishment, $N_p$ and $N_c$ are used to denote the control outputs in the prediction domain and the control domain, and the control outputs of Equation (16) in $N_p$ are expressed by

$$\left. \begin{array}{l} \boldsymbol{y}(k+1) = g(\boldsymbol{x}(k+1), \boldsymbol{u}(k+1)) \\ \boldsymbol{y}(k+2) = g(\boldsymbol{x}(k+2), \boldsymbol{u}(k+2)) \\ \vdots \\ \boldsymbol{y}(k+N_c) = g(\boldsymbol{x}(k+N_c), \boldsymbol{u}(k+N_c)) \\ \vdots \\ \boldsymbol{y}(k+N_p) = g(\boldsymbol{x}(k+N_p), \boldsymbol{u}(k+N_p)) \end{array} \right\}. \tag{19}$$

where the condition of $N_p$ and $N_c$ should satisfy $N_p \geq N_c$. In this paper, we set the control horizon equal to the prediction horizon, $N_p = N_c$, in all simulations and experiments in order to predict future UV states as accurately as possible.

Additionally, the system outputs $\boldsymbol{y}(k)$ and the control inputs $\boldsymbol{u}(k)$ can be given by

$$\boldsymbol{Y}(k+1) = \left[ \boldsymbol{y}(k+1), \boldsymbol{y}(k+2), \cdots, \boldsymbol{y}(k+N_p) \right]^T, \tag{20}$$

$$\boldsymbol{U}(k) = \left[ \boldsymbol{u}(k+1), \boldsymbol{u}(k+2), \cdots, \boldsymbol{u}(k+N_c) \right]^T. \tag{21}$$

The main objective of our expected NMPC design in $N_p$ is to minimize the tracking errors between the control output sequence and the reference trajectory sequence. To that end, the prescribed reference trajectory can be described as:

$$\boldsymbol{Y}_{ref}(k+1) = \left[ \boldsymbol{y}_{ref}(k+1), \boldsymbol{y}_{ref}(k+2), \cdots, \boldsymbol{y}_{ref}(k+N_p) \right]^T. \tag{22}$$

Herein, our main goal is to minimize the tracking error in the prediction time domain $N_p$, which is expressed by

$$min \| \boldsymbol{Y}(k+1) - \boldsymbol{Y}_{ref}(k+1) \|. \tag{23}$$

### 3.2. Formulation of Objective Function

In order to find out the minimized solutions to the tracking error system described in (23), it is necessary to further transform (23) into a linear-quadratic-regulator (LQR) control problem. To this end, the weighting factors $\boldsymbol{Q}$ and $\boldsymbol{R}$ are introduced and the following LQR optimization problem can be formulated, which is expressed by

$$J_{\text{cost}}(k) = \sum_{i=1}^{N_p} \left\{ \boldsymbol{y}(k+i) - \boldsymbol{y}_{ref}(k+i) \right\}^T \boldsymbol{Q} \left[ \boldsymbol{y}(k+i) - \boldsymbol{y}_{ref}(k+i) \right] + \\ \sum_{i=1}^{N_c} \left\{ \boldsymbol{u}(k+i) \right\}^T \boldsymbol{R} \boldsymbol{u}(k+i). \tag{24}$$

where $J_{\text{cost}}(k)$ is the optimized objective function, $i$ is the prediction step, $\boldsymbol{Q}$ is the weighting matrix that reveals the system's ability to follow up the reference road trajectory, $\boldsymbol{R}$ is the weighting matrix that manifests the stability performance of the trajectory tracking control problem, at time step $k$.

Therefore, the optimization formulation of the desirable NMPC controller can be expressed as

$$min \ J_{cost}(k). \tag{25}$$

Additionally, the physical constraints of the control input $u(k)$ described in (16) and (17) should be considered to find out the optimized solutions of (25). Finally, the nonlinear optimization problem of the NMPC controller design can be formulated as

$$\begin{aligned} x(k+i) = x(k+i-1) + f(x(k+i-1))T \\ + u(k+i-1), \end{aligned} \tag{26}$$

$$y(k+i-1) = g(x(k+i-1), u(k+i-1)), \tag{27}$$

$$u_{min}(k) \le u(k+i) \le u_{max}(k), \tag{28}$$

$$\Delta y_{min} \le y(k+i) - y_{ref}(k+i) \le \Delta y_{max}. \tag{29}$$

Practically, the optimal control sequence in $N_c$ at the sample time $k$ can be obtained as

$$u(k+i) = [u(k+1), u(k+2), \cdots u(k+N_c)]. \tag{30}$$

## 4. Simulation Verification

Here, comparative simulations have been performed under MATLAB software with the purpose of assessing the two MPC controllers: namely the traditional model predictive controller (TMPC) [27] and the proposed NMPC. The simulations are evaluated on this 2-DOF UV in the case of single-circle and twin-circle trajectory scenarios. Note that three indexes $x$, $y$, and $\varphi$ are used to demonstrate the performances of the two MPC controllers.

### 4.1. Simulations under a Straight-Line Trajectory

First, we expect this UV to move on a preset straight-line(ST) trajectory from the starting point (0, 0) at a longitudinal velocity $v = 1$ m/s. The mathematical expression of the reference straight trajectory is $Y_{ref} = 3$, and the simulation time was set as 5 s. Figure 3 displays the tracking response curves of this UV from the TMPC and the proposed NMPC controllers.

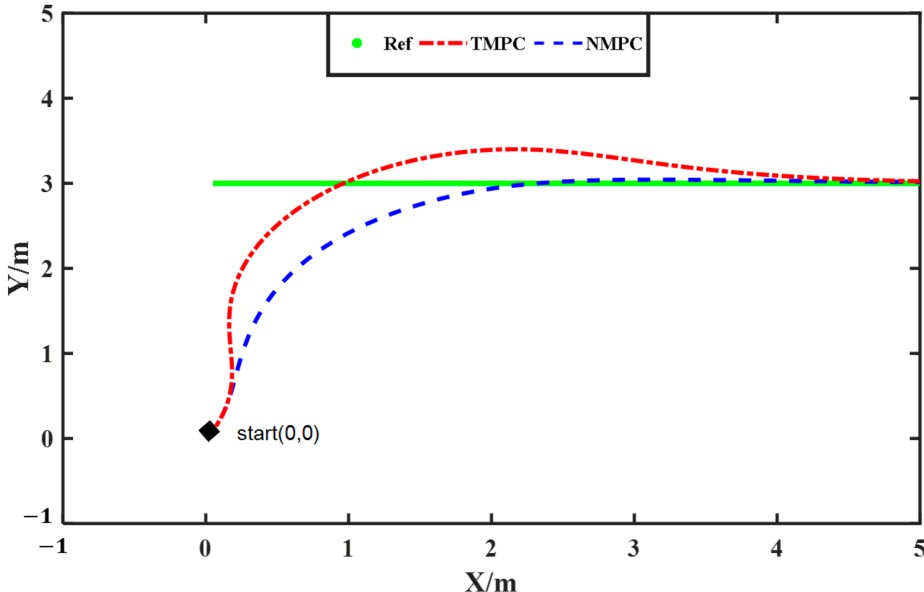

**Figure 3.** The simulation tracking results under the ST.

As shown in Figure 4, the three motion parameters $x$, $y$, and $\varphi$ of this UV followed overall smooth trends for the two controllers. Meanwhile, the three motion parameters $x$, $y$, and $\varphi$ converged to zero at about 2.5 s for our proposed NMPC and about 4.5 s for the TMPC.

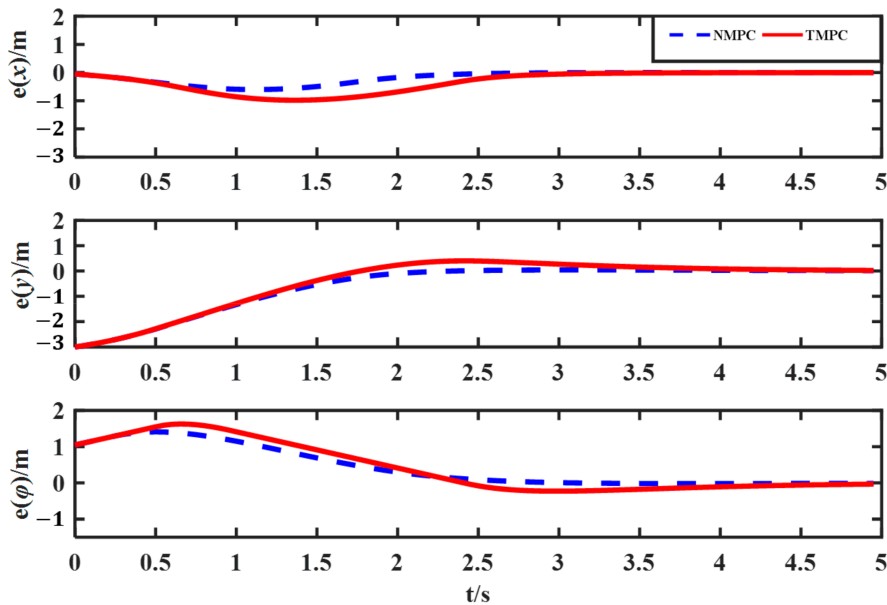

**Figure 4.** The tracking errors of $x$, $y$ and $\varphi$ under the ST.

Furthermore, the root mean square error (RMSE) value for the tracking errors of the three states is selected to quantitatively evaluate the control performance of the two MPC controllers, and based on the literature [28], the RMSE is defined by

$$\chi_{\text{RMSE}} = \frac{\|\chi_j - \chi_{ref,j}\|}{\sqrt{n}} = \sqrt{\frac{1}{n}\sum_{j=1}^{n}\left(\chi_j - \chi_{ref,j}\right)^2}, \; j = 1\ldots n. \tag{31}$$

wherein $\chi$ is an $n$-dimensional state vector, and $\chi_j$ is the value of $j$-th element in $\chi$, $\chi_{ref,j}$ is the value of $j$-th element in the reference road trajectory, $n$ is the length of $\chi$.

The results in Table 2 show that the RMSE values of $x$, $y$, and $\varphi$ for the proposed NMPC were reduced by about 51.14%, 1.35%, and 18.16% compared to those of the TMPC.

**Table 2.** The RMSE comparisons of the tracking errors of $x$, $y$, and $\varphi$ under the ST.

| Type | $x$ | $y$ | $\varphi$ |
|---|---|---|---|
| TMPC | 0.559 | 1.11 | 0.8264 |
| NMPC | 0.2731 (↓51.14%) | 1.095 (↓1.35%) | 0.6763 (↓18.16%) |

It can be seen from Figure 5 that under the ST condition, the change of the control input for the proposed NMPC is smoother than that of TMPC.

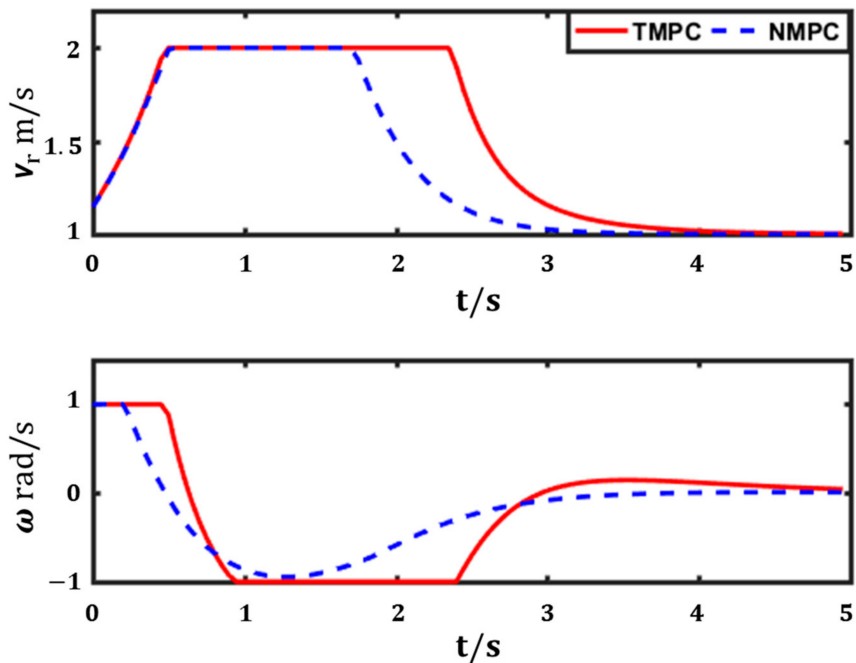

**Figure 5.** The control input of the TMPC and NMPC controllers under the ST.

*4.2. Simulations under a Single-Circle Trajectory*

A regular single-circle trajectory (SCT) road with a radius of 6 m is chosen as the reference trajectory road, and the corresponding simulations are conducted in this scenario. As shown in Figure 6, the simulation results generated by the TMPC and designed NMPC controller can well follow the prescribed reference road from the starting point $(-10, 0)$.

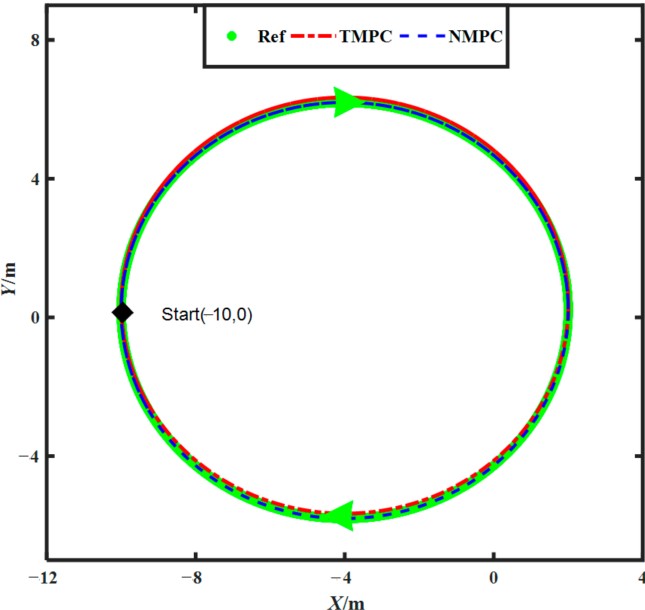

**Figure 6.** The simulation tracking results under the SCT.

Additionally, to examine the tracking accuracy of the two MPC controllers against the reference trajectory road, the tracking errors of $x$, $y$, and $\varphi$ for the UV under the SCT scenario are provided in Figure 7. It is obvious that the tracking errors of these three states of $x$, $y$, and $\varphi$ present a flatter trend with converging to zero. Moreover, in comparison with the TMPC controller, the tracking errors of $x$, $y$, and $\varphi$ for the proposed NMPC controller

have obviously fewer fluctuations on a whole and can reach a relative stability state in a shorter time. Particularly, the tracking errors of $x$, $y$, and $\varphi$ can converge to zero at about 2 s for the NMPC controller, and at 6 s for the TMPC controller.

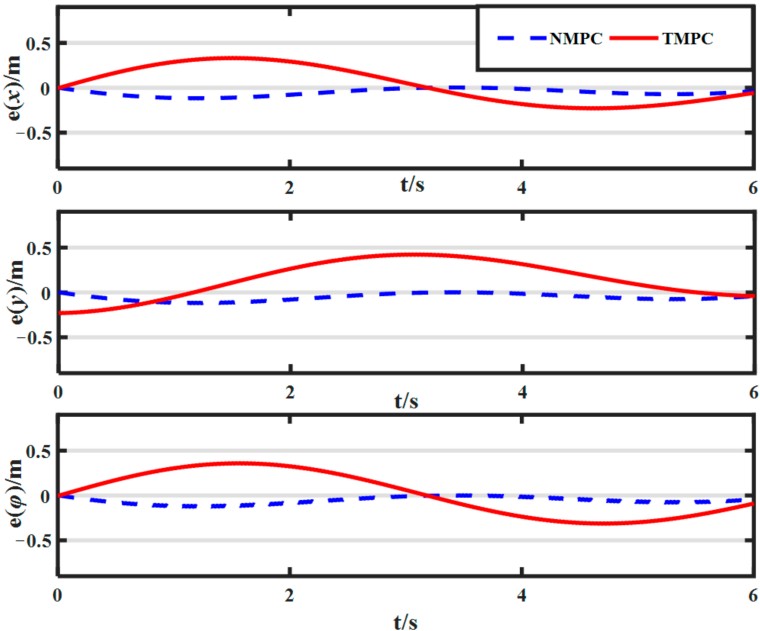

**Figure 7.** The tracking errors of $x$, $y$, and $\varphi$ under the SCT.

Herein, Table 3 lists the RMSE comparisons of the tracking errors of $x$, $y$, and $\varphi$ obtained by the TMPC and NMPC controllers under the SCT.

**Table 3.** The RMSE comparisons of the tracking errors of $x$, $y$, and $\varphi$ under the SCT.

| Type | $x$ | $y$ | $\varphi$ |
|---|---|---|---|
| TMPC | 0.2208 | 0.3734 | 0.2515 |
| NMPC | 0.1210 (↓45.20%) | 0.2284 (↓40.44%) | 0.1285 (↓48.95%) |

It can be seen from Figure 8 that the oscillation of the control input $v_r$ for the NMPC is much smaller than that of the TMPC under the SCT condition. Meanwhile, the control input $\omega$ of the NMPC can reach the target value faster.

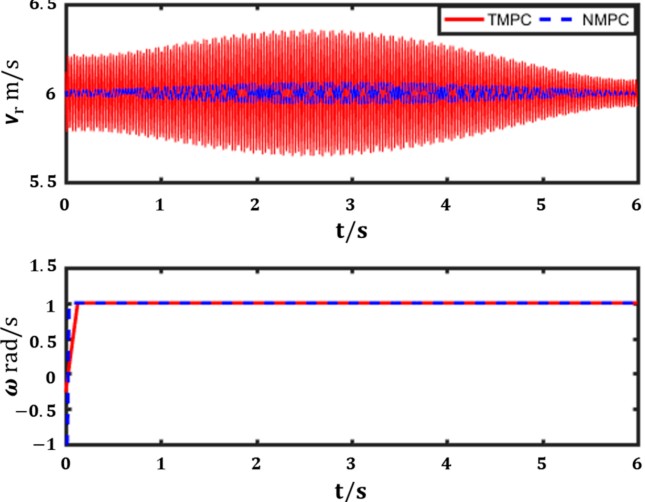

**Figure 8.** The control input of the TMPC and NMPC controllers under the SCT.

### 4.3. Simulations under a Twin-Circle Trajectory

To further evaluate the tracking performance of the NMPC controller, a twin-circle trajectory (TCT) road is used to conduct the simulations. The related tracking response curves are presented in Figure 9. It can be seen that both the TMPC and NMPC controllers can track the preset TCT starting from point $(-10, -0.2)$ closely. The UV is first running counterclockwise and then clockwise, after the lane change, our designed NMPC controller can still track another circular trajectory road with relatively higher accuracy. Furthermore, Figure 10 shows the tracking errors of $x$, $y$, and $\varphi$ for the two MPC controllers against the reference trajectory. It is obvious that the tracking errors of $x$, $y$, and $\varphi$ are relatively flat in this TCT scenario. In particular, the three states of $x$, $y$, and $\varphi$ can converge to zero at around 2 s for our presented NMPC controller, and 6 s for the TMPC controller.

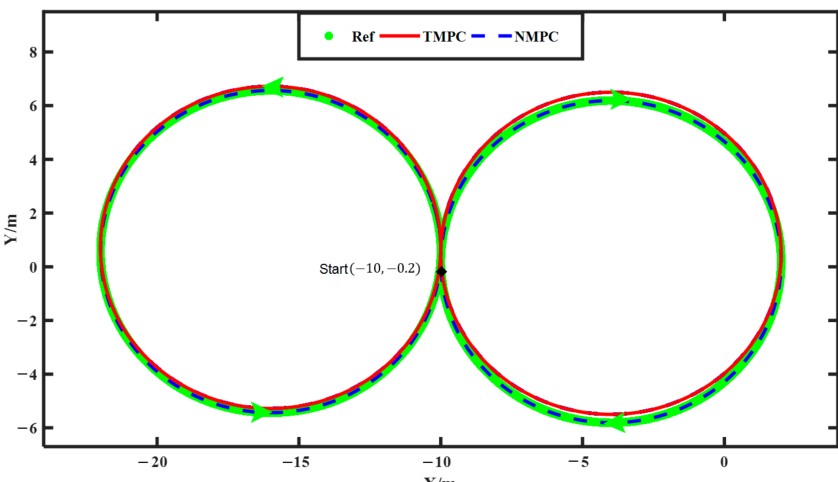

**Figure 9.** The simulation tracking results under the TCT.

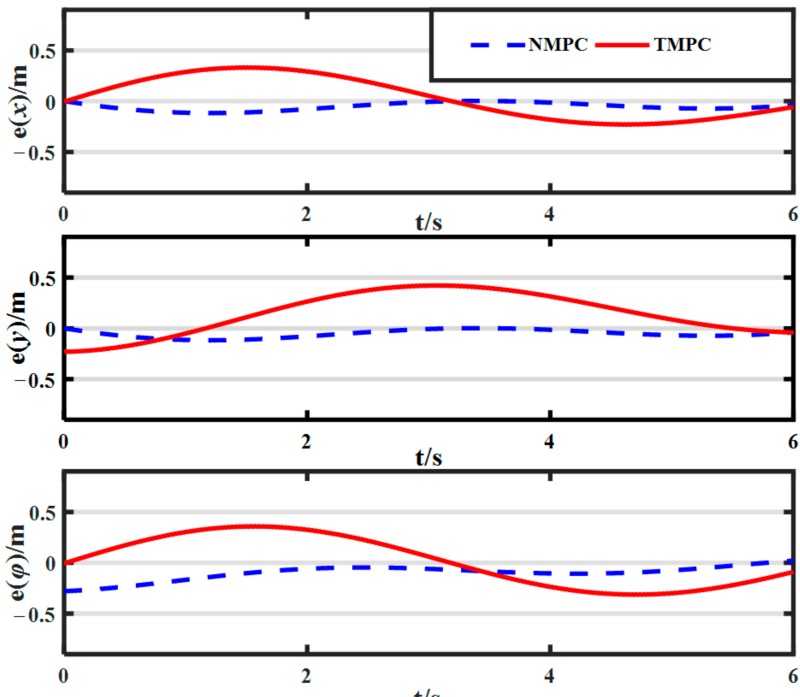

**Figure 10.** The tracking errors of $x$, $y$, and $\varphi$ under the TCT.

Table 4 summarizes the RMSE comparisons of the three states $x$, $y$, and $\varphi$ for the two MPC controllers under the TCT. Compared with the TMPC controller, the RMSE values of

$x$, $y$, and $\varphi$ for the presented NMPC can be reduced by about 65.35%, 54.61%, and 71.10%, respectively.

**Table 4.** The RMSE comparisons of the tracking errors of $x$, $y$, and $\varphi$ under the TCT.

| Type | $x$ | $y$ | $\varphi$ |
|---|---|---|---|
| TMPC | 0.1622 | 0.2743 | 0.1848 |
| NMPC | 0.0562 ($\downarrow$65.35%) | 0.1245 ($\downarrow$54.61%) | 0.0543 ($\downarrow$71.10%) |

It can be seen from Figure 11 that the control input $v_r$ for both MPC controllers have oscillations, but the oscillation amplitude of our proposed NMPC is significantly smaller than that of the TMPC. At the same time, the control input $\omega$ of the NMPC shows a faster response speed.

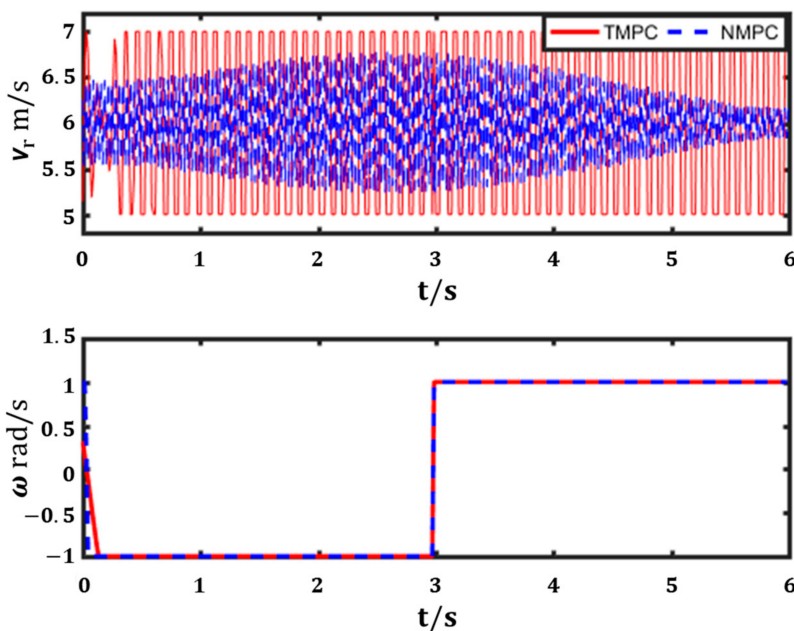

**Figure 11.** The control input of the TMPC and NMPC controllers under the TCT.

In general, our proposed NMPC controller exhibits better performance specifically for shorter tracking time and smaller fluctuation errors, in comparison with the TMPC controller. Additionally, according to the quantitative analysis of the tracking indexes as $x$, $y$, and $\varphi$, it is easily observed that our proposed NMPC controller has significant improvements in the tracking performance over the TMPC controller, which illustrates that the proposed NMPC has better control effects under three different driving scenarios.

## 5. Field Test Verification

In this section, an outdoor field test platform is constructed by ourselves in order to verify the effectiveness of those two MPC controllers. To facilitate the descriptions of this test platform, Figure 12 displays the schematic diagram of the experimental setup, and the outdoor field test photograph of this UV on the different prescribed road trajectories are provided in Figure 13. Different from the simulation verifications in Section 4, the field tests are conducted under two different driving scenarios.

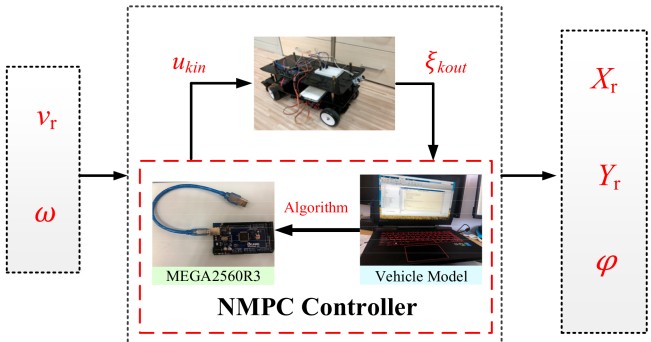

**Figure 12.** Schematic diagram of the experiment setup.

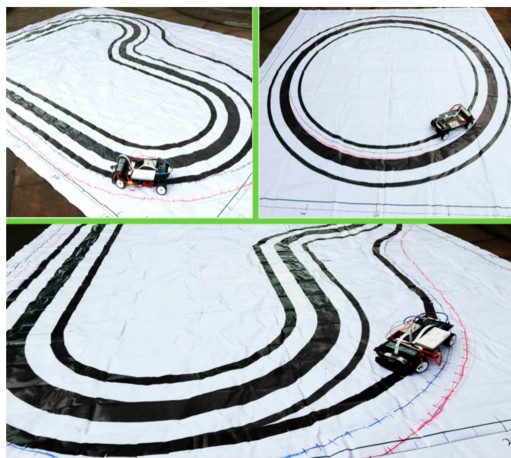

**Figure 13.** Outdoor field test setup.

Herein, a scaled unmanned vehicle—a BT-4 racing car, is used as the control plant, and an Arduino board (MEGA2560R3) is adopted to develop our proposed NMPC controller. Moreover, there are two DC (direct-current) motors in this BT-4 vehicle, one is used for steering, and another one is used to drive this UV at the rear wheel. Both DC motors are rooted in the Arduino (MEGA2560R3) board, using 54 digital I/O pins and 6-bit pulse width modulation (PWM) drivers. Because of the limitations of this self-established outdoor field test setup, only the measured points of $(x, y)$ at the ground coordinates are collected to draw the tracking trajectories for this BT-4 vehicle, which are further utilized to compare and validate the tracking performances of the two different MPC controllers.

*5.1. Test on Irregular Road*

For the practical driving scenario, the irregular trajectory road (ITR) is often encountered. Thus, the outdoor field tests are conducted on a practical ITR, and the tracking response curves of this UV are shown in Figure 14. In which, the solid green line denotes the reference trajectory road, the blue-dotted and the red-dashed lines denote the tracking response curves generated by the TMPC and NMPC controllers, respectively. Moreover, Figure 15 reveals the tracking errors of $x$ and $y$ for the two MPC controllers regarding the reference trajectory road. It should be noticed that this UV runs starting from the point $x_0 = \begin{bmatrix} 285 & 280 \end{bmatrix}$ at $v = 1$ m/s under the ITR.

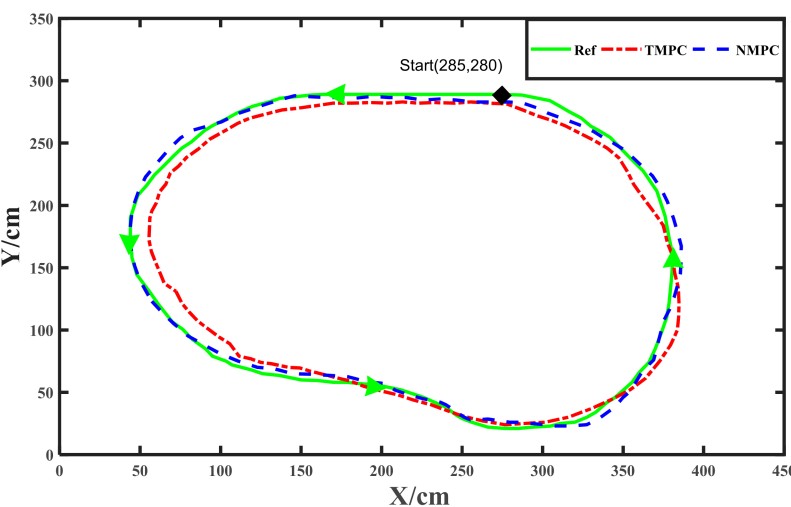

**Figure 14.** The tracking response curves of the two MPC controllers under the ITR.

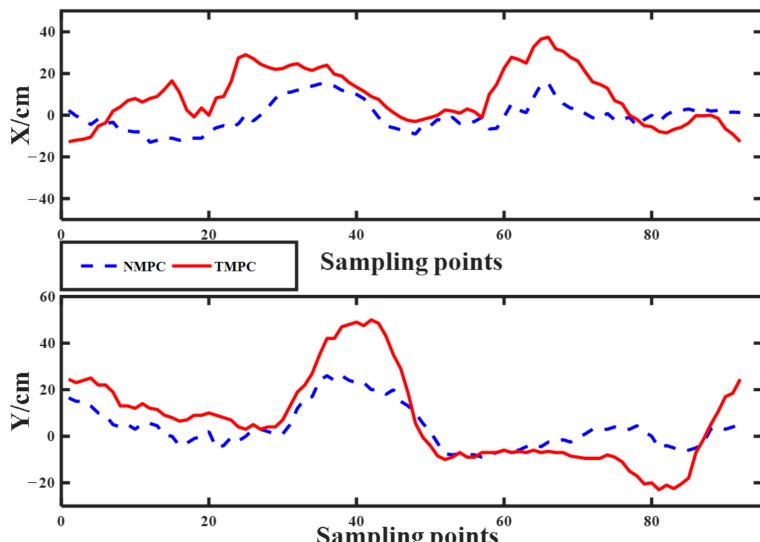

**Figure 15.** The tracking errors of $x$ and $y$ for the two MPC controllers under the ITR.

It is obvious from Figure 14 that the TMPC and proposed NMPC controller can track the ITR as possible as closely. Specifically, the blue-dashed line (denotes the tracking trajectory obtained by the NMPC controller) is much closer to the green-solid line (the reference trajectory) in comparison with the red-dashed line (denotes the tracking trajectory obtained by the TMPC controller). Furthermore, it is observed from Figure 15 that the tracking errors of $x$ and $y$ generated by the two MPC controllers could gradually converge to zero states, and the tracking error curves by the NMPC controller present a flatter appearance compared to the corresponding tracking curves by the TMPC controller. Particularly, the tracking error ranged from $-10$ cm to 15 cm for the NMPC controller and $-15$ cm to 35 cm for the TMPC controller in the X-direction. Additionally, in the Y-direction, the tracking errors of $x$ and $y$ are varied from 0 to 25 cm for the NMPC controller, and $-20$ cm to 50 cm for the TMPC controller, respectively.

Similar to the analysis of simulation verifications in Section 4, we herein use the RMSE comparisons of the tracking errors of $x$ and $y$ to evaluate the control performance of the designed controllers, which is presented in Table 5. In addition to this, the histogram comparisons of the tracking errors of $x$ and $y$ are also provided in Figure 16 to make a quantitative improvement of each tracking state for the TMPC and proposed NMPC controller under the ITR and the prescribed reference trajectory. Compared to the tracking

performance of the TMPC controller, the UV using the NMPC controller can reduce its RMSE values of $x$ and $y$ by about 54.72% and 48.65%, respectively.

**Table 5.** The RMSE of the tracking errors of $x$, $y$ under the single-circle road.

| Type | $x$ | $y$ |
|---|---|---|
| TMPC | 1.5800 | 2.0039 |
| NMPC | 0.7155 (↓54.72%) | 1.0029 (↓48.65%) |

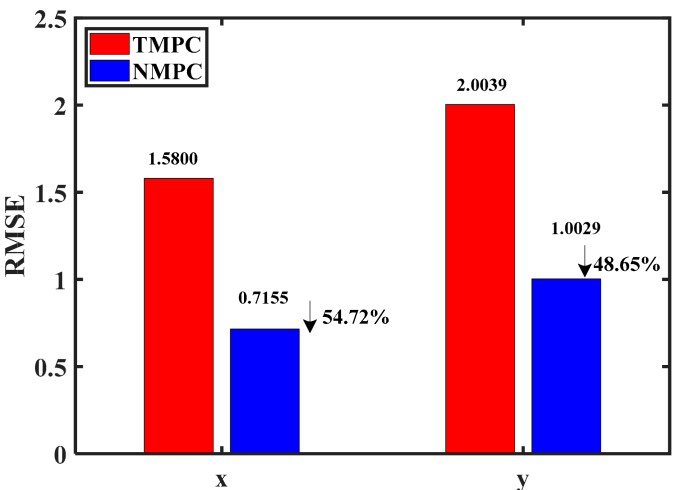

**Figure 16.** The histogram comparisons of two MPC controllers under the ITR.

### 5.2. Test on Double-Circle Road

Except for the irregular road trajectory, it is very necessary to validate the control performances of the proposed NMPC controller under a general double-circle trajectory road (DCTR). An outdoor field test is performed under a prescribed DCTR with a radius of 2.5 m, which starts from $x_0 = \begin{bmatrix} 250 & 50 \end{bmatrix}$ with $v = 1$ m/s. Figure 17 displays the tracking response curves obtained by the TMPC and NMPC controllers, and Figure 18 plots the tracking error curves of $x$ and $y$ for the two MPC controllers against the reference trajectory road.

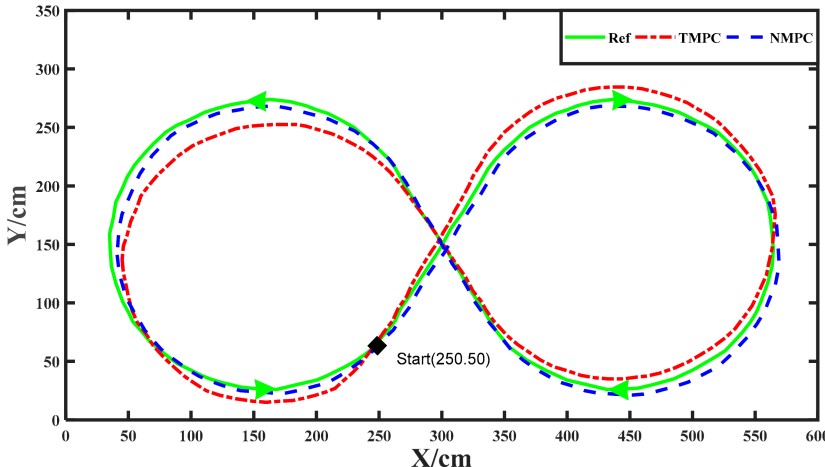

**Figure 17.** The tracking response curves of the two MPC controllers under the DCTR.

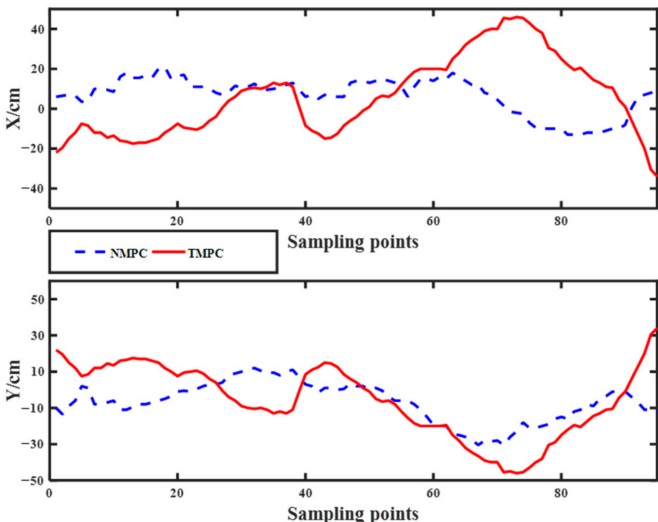

**Figure 18.** The tracking errors of $x$ and $y$ for the two MPC controllers under DCTR.

From Figure 17, it is easily seen that the two MPC controllers can nearly track the prescribed DCTR, and the tracking curves of the NMPC controller seem closer to the reference trajectory road. By a further observation from Figure 18, the tracking errors of $x$ and $y$ generated by the two MPC controllers are flat on a whole, yet the tracking errors by the NMPC controller can yield a smoother tendency and can enter into a relatively stable state at a faster rate compared to the TMPC controller. Besides, the tracking error of $x$ is varied from $-10$ to $15$ for the NMPC controller, and from $-30$ cm to $40$ cm for the TMPC controller in the X-direction. Moreover, the tracking error of $y$ is ranged from $-20$ cm to $10$ cm for the NMPC controller and from $-40$ cm to $30$ cm for the TMPC controller in the Y-direction.

Similarly, the RMSE comparisons of the tracking errors of $x$ and $y$ under this DCTR are quantitatively compared and provided in Table 6, and the histogram comparisons for the tracking errors of $x$ and $y$ are provided in Figure 16 to assess the control performance of the designed controller. It is clear from Table 6 and Figure 19 that the RMSE values of the tracking errors $x$ and $y$ for the NMPC controller are reduced by about 56.63% and 48.73%, respectively, in comparison with those of the TMPC controller, which further illustrates that the NMPC controller has better control effect under the DCTR scenario.

**Table 6.** The RMSE comparisons of the tracking errors of $x$ and $y$ under the DCTR.

| Type | $x$ | $y$ |
|------|-----|-----|
| TMPC | 2.7814 | 2.8215 |
| NMPC | 1.2061 (↓56.63%) | 1.4465 (↓48.73%) |

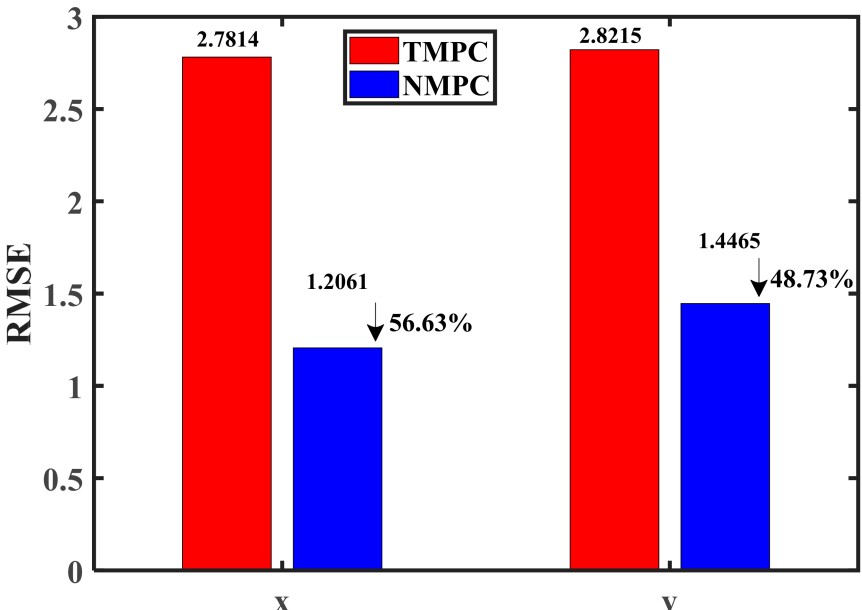

**Figure 19.** The histogram comparisons of two MPC controllers under the DCTR.

## 6. Conclusions

In this paper, a practical NMPC design method is proposed to achieve the accurate trajectory tracking application of a scaled UV. This desirable controller is developed based on a 2-DOF kinematics model of the UV, and in the framework of a standard MPC, the one-step Euler method is utilized to construct the nonlinear prediction model, then a nonlinear optimization objective function is formulated to minimize the tracking errors of forward velocity and yaw angle from a time-varying preset reference road. Finally, both the comparative simulations and the outdoor field tests are carried out to confirm the superior performances of the designed NMPC controller against the TMPC controller for this UV. The comparison of results demonstrates that the improvements of the tracking indexes $x$, $y$, and $\varphi$ for the UV with the presented NMPC controller are at least 45%, 1.35%, and 18.16%, respectively. Moreover, the field outdoor test results show that the improvements of tracking indexes $x$ and $y$ for the UV with the proposed NMPC controller are about 54% and 48%, respectively, compared with those with the TMPC controller. On a whole, it can be concluded that the NMPC controller outperforms better control performances regarding the TMPC controller.

Future study will focus on the trajectory tracking controller design of this UV considering the kinematics and dynamics properties simultaneously, and the experimental validation of the proposed controller in a real-time outdoor field test platform.

**Author Contributions:** Conceptualization, H.P.; methodology, H.P.; validation, M.L. and N.L.; formal analysis, M.L. and N.L.; resources, H.P. and C.H.; funding acquisition, H.P.; supervision, H.P. and C.H.; writing—original draft preparation, H.P.; writing—review and editing, H.P. and C.H. All authors have read and agreed to the published version of the manuscript.

**Funding:** This research was funded by the National Natural Science Foundation of China under Grant 51675423.

**Conflicts of Interest:** The authors declare no conflict of interest.

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
