# Peer review of "Practical Nonlinear Model Predictive Controller Design for Trajectory Tracking of Unmanned Vehicles"

_electronics, doi:10.3390/electronics11071110_

Round 1

Reviewer 1 Report

The paper introduces a non-linear MPC for path tracking control of an 4-wheeled vehicle driven by a servo motor installed in a rear-wheel and steered by a servo motor installed in a front-wheel. The MPC is derived from the non-linear dynamic model instead of a linearized model as in traditional MPC. The paper is very clear and easy to read.

The main weaknesses that can be improved:
- The controlled vehicle model is kind of simple and has been investigated quite widely in the litterature.
- The proposed non-linear MPC is compared with a traditional MPC, but the traditional one's derivation and parapemters are not given.
- In the validation simulations and experiments, the vehicle's starting point is always on the trajectory, and the trajectory is too simple (circles and smooth curves). Therefore, it is not possible to evaluate the controller performance in terms of stability, response time, overshoot, etc. The control inputs also need to be included.

Reviewer 2 Report

The presented results of experimental studies are the attractive part of the paper.

1. The references are mainly related to IEEE Transactions on Control Systems Technology and other IEEE Transactions. It seems not completely clear, why the Authors submitted their paper to Electronics instead of the mentioned journals. The Authors are encouraged to add some relevant references to this journal, demonstrating fitting the paper to the interests of the journal readers.

2. line 180 "According to the Shannon sampling theory" - the Authors should explain, how the Shannon sampling theory is applied to the significantly nonlinear model as (13), (14) is. The theoretical investigations concerning systems robustness with respect to sampling are known, but none of them is mentioned in the paper. What is more, the mentioned robustness (and in the linear case as well) is established with respect to the closed-loop system dynamics rather than to the open-loop controlled plant.

3. The explicit Euler method is used in the paper, although the implicit Euler scheme is known to be preferable. The authors are encouraged to consider this possibility too.

4. Reasoning about the accuracy of approximation of a nonlinear system by its discrete model (and the corresponding choice of the sampling step) is all the more relevant to a predictive model, for which the accuracy of the approximation is even more difficult to ensure. In the article, this issue is vaguely stated, including that the value of Nc is not indicated and its influence on the dynamics of the process as a whole is not examined.

Editing remarks.

1. line 108 "we can construct the 2-DOF kinematic model" - maybe "one can construct the 2-DOF kinematic model" would be better. line 122 - the same

2. lines 139-140 "Define... is further rewritten" - the sentence is not completely clear

3. line 176, Eq. (16) - meaning of "*" is not clear in the mathematical expression
line 220, Eq. (26) - the same

4. the use of square brackets to denote function arguments, and vectors as well, can confuse the reader. Usually, round brackets are used in the first case.

5. line 182 "For the control input ?(?), it cannot be arbitrarily changed, " - please reformulate

6. line 183 "physical constraints on ?(?), which is expressed as follows: " - "constraints ... is expressed" = please check

7. Eq. (18) imposes constraints on system output, not on control vector, as is stated above

Round 2

Reviewer 1 Report

The response to Reviewer 1's question 3 is not adequate:
(1) In the verification simulation and experiment, the smooth curve is adopted in the tracking map to consider the turning radius of the controlled vehicle.
- Smooth curves is useful only for evaluation of the steady state response. Transient response is also an important part in the evaluation of a controller's performance.

(2) The evaluation standard used in this paper is RMSE, which reflects the accuracy of the system tracking, and the system stability can also be observed from the error curve.
- RMSE is neccessary but not enough for an insight evaluation. Other useful metrics include response time, overshoot.

(3) The control input could not be collected due to the limitation of the experimental equipment.
- How about simulation at least?

Author Response

Please view the "electronics-Responses_R2_v1" file.

Reviewer 2 Report

The authors have improved the article in accordance with the comments and suggestions made, there are no more critical comments from my side.

Author Response

(The authors gave the same response as above.)

Round 3

Reviewer 1 Report

I recommend to accept the manuscript in the current form.